# An Intramolecular Hydroaminomethylation-Based Approach to Pyrrolizidine Alkaloids under Microwave-Assisted Heating

**DOI:** 10.3390/molecules27154762

**Published:** 2022-07-25

**Authors:** Elena Petricci, Simone Zurzolo, Camilla Matassini, Samuele Maramai, Francesca Cardona, Andrea Goti, Maurizio Taddei

**Affiliations:** 1Dipartimento di Biotecnologie, Chimica e Farmacia, Università degli studi di Siena, Via A. Moro 2, 53100 Siena, Italy; simone.zurzolo@student.unisi.it (S.Z.); maramai@unisi.it (S.M.); maurizio.taddei@unisi.it (M.T.); 2Dipartimento di Chimica “Ugo Schiff” (DICUS), Università degli studi di Firenze, Via della Lastruccia 13, 50019 Sesto Fiorentino, Italy; camilla.matassini@unifi.it (C.M.); francesca.cardona@unifi.it (F.C.); andrea.goti@unifi.it (A.G.)

**Keywords:** pyrrolizidine alkaloid, hydroaminomethylation, hydroformylation, microwave-assisted organic synthesis

## Abstract

A general method for the synthesis of pyrrolizidine derivatives using an intramolecular hydroaminomethylation protocol (HAM) under microwave (MW) dielectric heating is reported. Starting from a 3,4-bis(benzyloxy)-2-[(benzyloxy)methyl]-5-vinylpyrrolidine, MW-assisted intramolecular HAM in the presence of gaseous H2 and CO gave the natural alkaloid hyacinthacine A2 protected as benzyl ether. The same approach gave a lentiginosine analogue starting from the corresponding vinyl *N*-hydroxypyrrolidine. The nature of the reaction products and the yields were strongly influenced by the relative stereochemistry of the starting pyrrolidines, as well as by the catalyst/ligand employed. The use of ethanol as a solvent provides environmentally friendly conditions, while the ligand/catalyst system can be recovered by separating the alkaloid product with an SCX column and recycling the ethanolic solution. HAM worked up to three times with the recycled catalyst solution without any significant impact on yield.

## 1. Introduction

The pyrrolizidine nucleus can be considered as a privileged scaffold, since in nature more than 6000 animal or plant species hold pyrrolizidine alkaloids (Figure 1) [1,2], which are responsible for a wide range of biological activities depending on substituents and stereochemistry [3] Highly polyhydroxylated pyrrolizidine alkaloids can act as sugar mimics that inhibit glycosidases [4,5] The main pharmaceutical application of pyrrolizidine-based compounds is in traditional medicine (with the plant extracts used as ingredients), although several concerns have been posed regarding their safety on humans [3] In addition, some pyrrolizidine alkaloids (necine bases) and analogues can be used as pesticides thanks to their toxicity and natural deterrent effect on insects [6,7].

Among the various methods developed for their synthesis [8,9], the chiral pool strategy is one of the most effective approaches for the preparation of hyacinthacine A_2_ (**1**), alexine (**2**), of the natural indolizidine alkaloid lentiginosine (**4**) and pyrrolizidine analogues such as nor-lentiginosine (**3**). Natural proline, tartaric acid, and various sugars have been explored as enantiomerically pure starting materials, but the proposed multistep syntheses often suffer from lack of general applicability and in some cases scarce diastereoselectivity. Cycloadditions and sigmatropic rearrangements are probably the most used reactions for the synthesis of natural pyrrolizidines and their derivatives, along with condensations and metal-catalysed reactions [9]. A chemoenzymatic approach, starting from prolynal, has also been reported [10] to build up the polyhydroxylated part of the molecule (ring A). However, this method suffers from poor atomic economy, the use of non-environmentally friendly DMF and requires the separation of the diastereoisomers obtained. When sugars are used as starting material, the second cycle (ring B) is usually built up by ring-closing metathesis or amidation/alkylation reactions over the N atom, but variable stereoselectivity in positions 7a is commonly observed. The development of flexible and effective methods for the preparation of non-natural pyrrolizidine analogues is therefore an important topic, especially to investigate the structure-activity relationships of this class of molecules. Our previous studies have established nitrones derived from l-tartaric acid and d-arabinose or l-xylose [11] (**A** and **B** in Figure 1) as suitable substrates for efficient access to pyrrolizidine alkaloids by means of cycloaddition reactions [12] or additions of Grignard reagents followed by proper elaboration of adducts. Particularly, the addition of vinylMgBr to these nitrones occurred with nearly complete stereoselectivity[13] for steric and electronic reasons [14] and resulted amenable to the synthesis of hyacinthacine A_2_ [15] or nor-lentiginosine derivatives [16]after several synthetic steps.

We envisaged that a much more direct access to pyrrolizidine alkaloids such as hyacinthacine A_2_ (**1**) and nor-lentiginosine (**3**) might be provided by a hydroaminomethylation reaction (HAM) carried out on the same intermediates **C** [15] and **D** [17], respectively (Figure 1).[11,15,17,18,19,20,21,22,23,24] This could be an elegant approach for the preparation of polycyclic derivatives with promising large-scale applications, as the HAM process is based on the hydroformylation reaction (HF), the most commonly used catalytic transformation in the industry [25].

Although characterized by high atom economy, HF and HAM are featured by harsh reaction conditions in terms of temperature (>100 °C), syngas pressure (50 bar) and time (24 h) [26,27,28,29,30]. However, many reactions that require very harsh conditions, such as aminocarbonylation, hydrogen borrowing, reductive amination, HF, and HAM itself, can benefit from MW assistance [31,32,33,34,35,36,37,38,39,40,41,42,43,44,45]. We report here a generally relevant, mild, and sustainable method for the synthesis of pyrrolizidine nuclei via an intramolecular HAM process under MW dielectric heating.

## 2. Results and Discussion

Our retrosynthetic strategy for the synthesis of generic pyrrolizidine derivatives is shown in Figure 2. The pyrrolizidine skeleton can be obtained by the HAM of olefin **E** or **F** derived from the reduction of hydroxylamine **C** or **D**, which in turn was obtained by the stereoselective addition of vinylmagnesium bromide to enantiomerically pure nitrones **A** or **B**.

In the intramolecular HAM process, we expected the aldehyde to form an iminium/enamine intermediate after HF on the olefin [28,29,46]. Nucleophilic binding of the amine, followed by reduction catalyzed by the Rh catalysts, would give the expected bicyclic pyrrolizidine product. To set up the process, we investigated the use of both hydroxylamine **C** and **D** and the corresponding amine **E** and **F**, possibly formed by the in-situ reduction in the reaction conditions.

We started from nitrone **5** [47], an intermediate suitable for the synthesis of the nor-lentiginosine (**3**). A highly diastereoselective addition of vinylmagnesium bromide 6 gave hydroxylamine **7** in 96% yield and allowed the incorporation of a new (*S*)-configured stereogenic center in C-2 position (Figure 3) [17]. Compound **7** and the corresponding amine **9**, obtained by reduction with catalytic In and Zn in NH_4_Cl solution [48,49], served as model substrates for the HAM process under MW dielectric heating. Different conditions were tested for the reactions, varying the catalyst/ligand, temperature, and solvent used, and the results are shown in Table 1.

Under standard MW-assisted conditions for HF (H_2_/CO, 1/1, Xantphos/(PPh_3_)_3_Rh(CO)H in toluene containing [bmim][BF_4_], 110 °C, 30 min) [45], **7** gave the aldehyde 10 as the main reaction product (44%) together with a **10**% of the desired bicyclic product **12** and traces of enamine **11** (Table 1, entry 1).

Under the same reaction conditions, the amine **9** furnished solely the enamine **11**, but with a poor 27% conversion. Better results were obtained with EtOH as solvent, especially starting from hydroxylamine **7** that gave the aldehyde **8a**, in equilibrium with its cyclic hemiacetal form **8b**, as the major reaction product in 65% isolated yield, while **9** showed almost no reactivity (Table 1, entry 2). In toluene alone, at lower temperatures (80 °C), no conversion was observed starting from **7**, while starting from **9** only the hydrogenation by-product **13b** was formed (Table 1, entry 3). At higher temperatures (130 °C), a relatively small amount of the enamine **11** was obtained from the two vinylpyrrolidines **7** and **9** (Table 1, entry 4). HAM with Biphephos or PPh_3_ as ligands resulted in the hydrogenation of the C=C double bond and product **13** was obtained as the major component of the reaction mixture (40–90%) regardless of the starting material (Table 1, entries 5–6). Other variations in conditions, such as [RhCl(COD)]_2_ as catalysts, pressure of syngas (3–14 bar) and reaction times (40–60 min) were tested without improvement in conversion or selectivity. The better result was then the conversion of hydroxylamine **7** into the mixture of **8a** and **8b** using EtOH as solvent. Treatment of this mixture with PTSA in refluxing toluene (30 min, MW) reversed the equilibrium towards aldehyde **8a**, which could be hydrogenated (H_2_, 1 bar, MeOH MW, 40 °C, 15 min) to give compound **12** that, upon deprotection with trifluoroacetic acid, ref. [17] afforded nor-lentiginosine (**3**) (Figure 4).

However, when we explored the possible application of this methodology to a differently protected 2-vinyl-***N***-hydroxypyrrolidine hydroxylamine (**14**, Figure 5), we were able to obtain only the intermediate aldehyde mixture **15a/15b** in just 37% isolated yields. The following transformation into the corresponding bicyclic derivative **16** by reducing the hydroxylamine and subjecting the product to intramolecular reductive amination conditions did not occur using H_2_ and Pd/C under both traditional heating or MW dielectric heating.

For the synthesis of hyacinthacine A_2_ (**1**), we started from the known hydroxylamine **17** obtained from the corresponding nitrone and vinylmagnesium bromide [15]. Under the previously established conditions for HF, 7 bar syngas in the presence of Xantphos/(PPh_3_)_3_Rh(CO)H in toluene and [bmim][BF_4_] at 110 °C for 30 min, only a small amount of hydrogenation product **18** was obtained (Table 2, entry 1), while most of the starting material remained unreacted. Increasing the temperature to 130 °C resulted in only 27% conversion to the desired compound **22** together with 2% of the enamine **21**, but 56% of the starting material **17** was still present in the final reaction mixture (Figure 6 and Table 2, entry 2).

With EtOH as solvent instead of toluene, compound **17** did not react at all (Table 2, entry 3). Increasing the syngas pressure had no effect, while using Biphephos as a ligand gave a complete conversion to the hydrogenation compound **18a** (Table 2, entry 4). Pyrrolidine **19** was then prepared following our well-established protocol that employs catalytic indium and stoichiometric Zn as reducing agent [48,50], and HAM was investigated under different conditions (Figure 5). The unsaturated pyrrolizidine **21** was isolated from **19** in toluene at 110 °C in 43 % yield (Table 2, entry 5). At higher temperature (130 °C), a mixture of products **20**, **21**, and **22** was observed (Table 2, entry 6). However, the use of EtOH as a solvent allowed almost complete conversion of **19** into **22** (Table 2, entry 7). When different ligands (Biphephos, PPh_3_) were used, a reduction of the C=C bond with a concomitant reduction of the N-O bond was the only reaction observed (Table 2, entries 8 and 9). Lower or higher syngas pressures always led to worse results in terms of yield of compound **22**. Hydrogenolysis of **21** or **22** over Pd/C has been previously reported to give hyacinthacine A_2_ (**1**) [15,51,52,53] and is here used together with MW irradiation. Moreover, as we worked in a very small scale, the final product was obtained in such a low amount that good quality NMR spectra could not be recorded. Furthermore, both elemental analysis and [α]_D_ are in agreement with data reported in the literature.

To investigate the possibility of reusing both the catalyst and the solvent used, the reaction was carried out on a 100-mg scale and the crude product filtered directly through an SCX column. Pyrrolizidine **22** was retained by the column and further recovered by washing the column with NH_3_ in EtOH. The first ethanolic fraction containing the ligand and catalyst was then recycled by addition of amine **19** and the solution was subjected to a new HAM cycle under the conditions given in entry 7 of Table 2. The expected product **22** was obtained with 80% yield. The recycle was repeated 3 times without any effect on the reaction yield, proving that the catalyst can be recycled efficiently. It is worth noting that, for all substrates studied, the CO/H_2_ addition to the C=C bond occurs regioselectively at the terminal position, as no branched aldehydes or the corresponding hemiacetals were observed.

In summary, we have shown that intramolecular HAM is a transformation that can be applied to the synthesis of pyrrolizidine alkaloids and their derivatives with high atom economy. We investigated both the use of hydroxylamines obtained from the reaction of nitrones with vinylmagnesium bromide and the corresponding amines. The use of the hydroxylamine as a substrate for HAM gave an intermediate aldehyde that was directly reduced to pyrrolizidine derivatives. In our experiments with different substrates, we found that the decoration of the A ring and/or the configuration at the different stereogenic centres have an influence on the reaction outcome. The hydroxylamine proved to be effective for our purposes only in the synthesis of the nor-lentiginosine **3** by a two-step process. Indeed, the intermediate aldehyde (**8a**) formed by HF remains in the corresponding unreactive hemiacetal form (**8b**), which requires further treatment and subsequent reducing conditions to afford **3**. However, once a substituent was introduced at C-3 of the pyrrolidine ring, the amine performed better than the corresponding hydroxylamine and afforded the pyrrolizidine derivative in a one-pot reaction with 82% isolated yield. In all cases, intermediate enamine derivatives, potentially useful for other synthetic aims, were formed in different ratios, depending on the reaction conditions and catalyst/ligand used.

## 3. Materials and Methods

All reagents were used as purchased from commercial suppliers without further purification. Flash column chromatography was performed in glass columns using Merk silica gel 60 Å, 230–400 mesh particle size. For analytical thin-layer chromatography, Merck aluminium-backed plates pre-coated with silica gel 60 (UV254) were used and visualised by staining with a solution of p-anisaldehyde in EtOH or a KMnO_4_ solution. ^1^H NMR and ^13^C NMR spectra were recorded using a 400 MHz Brucker Advance NMR spectrometer (Bruker BioSpin AG, Fällanden, Switzerland). Deuterated chloroform and methanol were used as solvents. Chemical shift (δ) values are given in parts per million (ppm) and refer to the residual signals of the deuterated solvent (δ 7.26 for 1H and δ 77.6 for ^13^C in CDCl_3_, δ 3.34 for ^1^H and δ 49.00 for ^13^C in CD_3_OD). The data are presented as follows: chemical shift, multiplicity (s = singlet, d = doublet, dd = doublet of doublets, dt = doublet of triplets, t = triplet, q = quartet, m = multiplet or multiple resonances, bs = broad singlet), coupling constant (J) in hertz and the integration in ppm. Mass spectrometry data were collected using an Agilent 1100 LC /MSD VL system.

MW assisted reactions are performed in a CEM Discover MW (CEM s.r.l., Cologno Al Serio (BG), Italy) oven equipped with a 10 mL tube for reactions under pressure and an external IR sensor to record the reaction temperature during irradiation (CEM s.r.l., Cologno Al Serio (BG), Italy). This glass vial, tested to withstand pressures up to 250 psi (17 bar, 1723 KPa), is equipped with a tubing connection to an external pressure control system that includes a valve and output tubing to vent the vial at the end of the reaction. The output tubing was connected via a three-way connector to a cylinder containing CO/H_2_ (1:1) equipped with two taps to pre-purge the system prior to MW irradiation.

The nitrone derivatives were synthesised as previously reported [47,51,54].

General method for the preparation of 1-hydroxy-2-vynylpyrrolidine derivatives **7*,* 14*,* 17** [17]: To a stirred solution of the proper nitrone (4.37 mmol) in dry Et_2_O (40 mL), a 1 M solution of vinyl magnesium bromide in THF (5.5 mL, 5.5 mmol), was slowly added under nitrogen atmosphere at 20 °C. After stirring at 20 °C for 1 h and 45 min, 15 mL of saturated aqueous NaHCO_3_ were added. The precipitate was filtered and the mixture extracted with diethyl ether (3 × 15 mL). The combined organic extracts were dried over dry Na_2_SO_4_, and the solvent evaporated under vacuum and directly used in the next step. Single ^1^H-NMR spectra in Appendix A.

General procedure for the synthesis of 2-vinylpirrolidine **9** and **19** [17]: To a stirred solution of the proper 1-hydroxy-2vinylpirrolidine (0.83 mmol) in dry methanol (10 mL), a saturated solution of NH_4_Cl (15 mL), powdered Zn (218 mg, 3.33 mmol) and a catalytic amount of indium dust (1.7 mg, 0.015 mmol) were added at 20 °C. The mixture was reflux overnight under N_2_. The solvent was evaporated under vacuum and a saturated aqueous solution of Na_2_CO_3_ (15 mL) was added. The mixture was extracted with Et_2_O (3 × 5 mL) and the combined organic phases were dried over dry Na_2_SO_4_, and the solvent evaporated under ***vacuum***. The brownish oil (was used in the next step. Single ^1^H-NMR spectra in Appendix A.

**3-[(2*S*,3*S*,4*S*)-3,4-Di-*tert*-butoxy-1-hydroxy-2-pyrrolidinyl]propionaldehyde (8a)**: To a solution of **7** (30 mg, 0.16 mmol) in EtOH (500 μL), (PPh_3_)_3_Rh(CO)H (18 mg, 0.02 mmol) and Xantphos (46 mg, 0.08 mmol) were added. The yellow solution obtained was submitted to pressurized syngas at 100 psi (7 bar) and heated for 30 min at 110 °C by MW dielectric heating at 150 W (value previously settled on the MW oven). The flask was cooled down to rt and the internal gas released. The reaction mixture was evaporated in vacuo and the yellow oil obtained was purified by flash chromatography (CHCl_3_/MeOH: 95/5). Expected product **8** was obtained as a pale brown oil in 65% yields. ES-MS: 288 [M + H]^+^, 310 [M + Na]^+^. ^1^H-NMR (400 MHz, CDCl_3_): δ 9.77 (s, 1H, C***H***O), 4.37–4.25 (m, 1H, 4-***H***), 3.94–3.87 (m, 1H, 3-***H***), 3.17–3.11 (m, 1H, 5-***H***a), 3.00 (t, J = 8.28 Hz, 1H, 5-***H***b), 2.76–2.66 (m, 1H, 2-***H***), 2.46–20142.28 (m, 4H, C***H***_2_C***H***_2_), 1.25 (s, 9H, C***H***_3_ × 3), 1.23 (s, 9H, C***H***_3_ × 3) ppm. ^13^C-NMR (100 MHz, CDCl_3_, 308.15 K): δ 203.8, 81.4, 78.6, 77.8, 77.1, 41.1, 28.9, 19.7 ppm. Elemental Analysis for C_15_H_29_NO_4_: calcd. C-62.69, H-10.17, N-4.87, O-22.27; found C-62.45; H-10.43; N-4.92.

**(1*S*,2*S*,7a*S*)-Hexahydro-1*H*-pyrrolizine-1,2-diol (3):** A solution of **8** (16 mg, 0.06 mmol) in toluene (1 mL) in the presence of ***p***-toluenesulfonic acid (1 mg, 0.0056 mmol) was submitted to MWs in open vessel conditions at 120 °C for 30 min at 300 W (value previously settled on the MW oven). The reaction mixture was evaporated in vacuo and the oil obtained was dissolved in MeOH (1 mL). Pd/C 10% wt (5.9 mg, 0.0056 mmol), and HCl 12 N (1 μL) was added and suspension obtained was submitted to two consecutive vacuum/H_2_ cycles, then pressurized with 15 psi (1 bar) of H_2_ and heated for 30 min at 40 °C by MW dielectric heating at 50 W (value previously settled on the MW oven). The flask was cooled to rt and the internal gas released. The reaction mixture was filtered over a Celite pad with MeOH (2 × 5 mL) and the solution passed through an SCX column washing with EtOH (2 × 10 mL). The column was washed with a 30% *v/v* NH_3_ solution in EtOH (2 × 10 mL): the solution obtained was evaporated in vacuo obtaining **12** that was directly treated with a 10% solution of CF_3_COOH in CH_2_Cl_2_ (2 mL) at r.t. for 3 h. The mixture was treated with a solution of NH_3_ in EtOH and passed through an SCX colum. After evaporation in vacuo **3** was obtained (4 mg) in 40 % yield. ES-MS: 144 [M + H]^+^, 166 [M + Na]^+^, 309 [2M + Na]^+^. ^1^H-NMR (400 MHz, D_2_O): δ 4.21 (q, J = 5.3 Hz, 1H), 3.92–3.99 (m, 1H), 3.48–3.66 (m, 2H), 3.24–3.32 (m, 1H), 2.83-3.01 (m, 2H), 1.73–17 (m, 4H) ppm. ^13^C-NMR (100 MHz, D_2_O): δ 82.09, 77.88, 71.74, 58.89, 58.03, 29.57, 26.27 ppm. Elemental Analysis for C_7_H_13_NO_2_: calcd. C-58.72; H-9.15; N-9.78; O-22.35; found C-58.37; H-9.24; N-9.83.

**(1*R*,2*R*,3*R*,7a*R*)-1,2-Bis(benzyloxy)-3-[(benzyloxy)methyl]hexahydro-1*H*-pyrrolizine (22):** To a solution of **19** (30 mg, 0.067 mmol) in EtOH (500 μL), (PPh_3_)_3_Rh(CO)H (18 mg, 0.02 mmol) and Xantphos (46 mg, 0.08 mmol) were added. The yellow solution obtained was submitted to pressurized syngas at 100 psi (7 bar) and heated for 30 min at 110 °C by MW dielectric heating at 150 W (value previously settled on the MW oven). The flask was cooled to rt and the internal gas released. The reaction mixture was passed through an SCX column washing with EtOH (2 × 10 mL). The column was then washed with a 30% ***v***/***v*** NH_3_ solution in EtOH (2 × 10 mL): the solution obtained was evaporated in vacuo obtaining **2** (24 mg) as a pale yellow solid in 82 % yield. M.p.: 49–51 °C Lit. [52] 47.5 °C ES-MS: 444 [M+H]^+^. ^1^H-NMR (400 MHz, CDCl_3_): δ 7.30–7.25 (m, 15H, Ar***H***), 4.66 (d, J = 11.6 Hz, 1H, OC***H***_2_Ph); 4.67–4.49 (m, 5H, OC***H***_2_Ph), 4.04 (t, J = 7.2 Hz, 1H, 3-C***H***), 3.78 (t, J = 7.2 Hz, 1H, 3-C***H***), 3.56–3.45 (m, 3H, 1-***H***, 2-***H***, 7a-***H***), 3.10–3.03 (m, 1H, 3-C***H***), 3.00–2.92 (m, 1H, 5-C***H***), 2.81**–**2.72 (m, 1H, 5-***H***), 1.99**–**1.57 (m, 4H, 6-C***H***_2_, 7-CH_2_) ppm. [α]^25^_D_ -4.7 (***c*** 1, CHCl_3_) Lit. [52] [α]^25^_D_ -5 (***c*** 1, CHCl_3_). Lit. [21] [α]^24^_D_ -5.1 (***c*** 0.6, CHCl_3_). Elemental Analysis for C_29_H_33_NO_3_: calcd. C-78.52; H-7.50; N-3.16; O-10.82; found C-78.79; H-7.34; N-3.02.

**(1*R*,2*R*,3*R*,7a*R*)-3-(Hydroxymethyl)hexahydro-1*H*-pyrrolizine-1,2-diol (1):** A suspension of **17** (24 mg, 0.05 mmol), Pd/C 10% wt (5.3 mg, 0.005 mmol), and HCl 12 N (3 μL) in MeOH was submitted to 2 vacuum/H_2_ cycles, then pressurized with 15 psi (1 bar) of H_2_, and heated for 30 min at 80 °C by MW irradiation at 50W (value previously settled on the MW oven). The flask was cooled down to r.t. and the internal gas released. The reaction mixture was filtered over a Celite pad with MeOH (2 × 5 mL) and the solution passed through an SCX column washing with EtOH (2 × 10 mL). The column was washed with a 30% *v/v* NH_3_ solution in EtOH (2 × 10 mL): the solution obtained was evaporated in vacuo obtaining **1** in quantitative yields. [α]^25^_D_ +12.1 (***c*** 0.4, D_2_O) Lit. [53] [α]^25^_D_ +12.5 (***c*** 0.4, H_2_O). Lit. [21] [α]^24^_D_ +12.4 (***c*** 0.2, H_2_O). Elemental Analysis for C_8_H_15_NO_3_: calcd. C-55.47, H-8.73, N-8.09, O-27.71; found C 55.58, H 7.87, N 8.38.

## Data Availability

Not applicable.

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
