# Peer review of "An Intramolecular Hydroaminomethylation-Based Approach to Pyrrolizidine Alkaloids under Microwave-Assisted Heating"

_molecules, 2022, doi:10.3390/molecules27154762_

Round 1

Reviewer 1 Report

The paper from Patricci and col. describes a complete study on a hydroformilation reaction under microwave irradiation to be used in the preparation of pyrrolizinide alkaloids.

I think the paper is interesting enough to be published with only some minor corrections.

- Graph 1 seems to be very clear but it is very confusing for my. I think it could be more simple to replace it by a table with the number of compounds and yields.

- It seems that MW reactions are in temperature controlled conditions. If this is the case the power (50-300 W) is only the maximum power used. This should be clarified in the text. 

- In open vessel conditions, they use a reflux condenser?

Author Response

- Graph 1 has been substituted by more clear Table 1.

- Indication of the conditions in term of power used have been reported in table 1 note [a]. The power is only the maximum settled to reach the expected temperature

- In open vessel conditions indicate the use of a reflux condenser

Reviewer 2 Report

E. Petricci and co-workers describe the synthesis of some pyrrolizidine-based compounds via an hydroaminomethylation using microwaves as heat source. Thus, I found enough positive features that make this manuscript publishable in Molecules, for instance, enough novelty and originality come from this work, synthesized products are of interest in medicinal chemistry, good stereoselectivity was observed, introduction shows suitably the context of the chemistry developed, references cited are enough and pertinent, discussion of results is okay, green conditions like using EtOH and recovery-recyclability of catalysts were well stablished, and experimental section seems to be fine. However, one major point together with a few minor aspects must be attended before publication, as follows:

Major point

- Nowadays, it is mandatory to submit an Electronic Supplementary Material (ESI) along with the manuscript, to include spectra (MS, IR and NMR), especially of all new products like 8a, 3, 22, 1, and so on. Authors should not have any problem to do it.

Minor aspects  

- I would modify a bit the tittle by “An intramolecular hydroaminomethylation-based approach to pyrrolizidine alkaloids under microwave-assisted heating”

- The Figure 2 is so confusing. There are repeated labels ‘previous work’ and ´This work’, which may cause misunderstandings. At least for me it was not clear what arrows are for “Previous work” and what ones for “This work”.

- According to MDPI instruction for authors, all artworks (Figures, Schemes, Tables, etc.) must have captions

- With respect to Graph 1, I would place the yield above all colored section for each bar to prevent reader trying to guess those values. In this context, 37 percentages must be included there (Graph 1). In the same way, replace SM by 9. Also in Tab 1, replace SM for 17.

Author Response

Major point

- Nowadays, it is mandatory to submit an Electronic Supplementary Material (ESI) along with the manuscript, to include spectra (MS, IR and NMR), especially of all new products like 8a, 3, 22, 1, and so on. Authors should not have any problem to do it. DONE

Minor aspects  

- I would modify a bit the tittle by “An intramolecular hydroaminomethylation-based approach to pyrrolizidine alkaloids under microwave-assisted heating” DONE

- The Figure 2 is so confusing. There are repeated labels ‘previous work’ and ´This work’, which may cause misunderstandings. At least for me it was not clear what arrows are for “Previous work” and what ones for “This work”. DONE

- According to MDPI instruction for authors, all artworks (Figures, Schemes, Tables, etc.) must have captions DONE

- With respect to Graph 1, I would place the yield above all colored section for each bar to prevent reader trying to guess those values. In this context, 37 percentages must be included there (Graph 1). In the same way, replace SM by 9. Also in Tab 1, replace SM for 17. DONE

Reviewer 3 Report

This work describes an apparently new procedure for the preparation of haycinthacine A2 and nor-lentiginosine. The key step for the preparation of the target compounds is the hydroformylation reaction. The results of this work are interesting from a chemical point of view, however, their presentation requires dramatic improvement.

For example: all figures and schemes lack a caption. 

Figure 2 is rather scheme than figure. There it is not quite clear what the goal of the work is (Scheme is complex and "This work" blends in with the rest of scheme)

The text on page 2, line 61-64 refers to Figure 2, where compounds C and D should be listed, but they are not.

The text on page 3, line 85-90 would merit a mechanistic scheme, as it is not clear from the text what the authors mean.

page 3, line 94: Where is the stereogenic centre C-7a? For better clarity, I recommend to show it in scheme.

How was the structure of substance 7 confirmed? Moreover, an ESI file is not attached to the paper. Therefore, it is difficult to judge the correctness of the presented results.

The results presented in Graph 1 are not clear. Yields are given in the text, but these cannot be obtained from the graph.

How was substance 14 prepared?

In Scheme 5, the synthesis of compound 1 is given by reference. This gives the impression that it is a formal synthesis, but this is not true. Complete the reaction conditions and yield of compound 1.

Page 7, line 217: What nitrones were prepared as previously reported? Please add. 

Author Response

For example: all figures and schemes lack a caption. DONE

Figure 2 is rather scheme than figure. There it is not quite clear what the goal of the work is (Scheme is complex and "This work" blends in with the rest of scheme) DONE

The text on page 2, line 61-64 refers to Figure 2, where compounds C and D should be listed, but they are not. DONE

The text on page 3, line 85-90 would merit a mechanistic scheme, as it is not clear from the text what the authors mean. Introduce in Scheme 2

page 3, line 94: Where is the stereogenic centre C-7a? For better clarity, I recommend to show it in scheme. Is reported in Scheme 1B

How was the structure of substance 7 confirmed? Moreover, an ESI file is not attached to the paper. Therefore, it is difficult to judge the correctness of the presented results. 7 was already published by some of us and we confirm the structure by NMR, and alfa-D. 1-NMR is reported in SI.

The results presented in Graph 1 are not clear. Yields are given in the text, but these cannot be obtained from the graph. Table 1 replace Graph 1

How was substance 14 prepared? Reported in Scheme 4 and NMR spectra in SI

In Scheme 5, the synthesis of compound 1 is given by reference. This gives the impression that it is a formal synthesis, but this is not true. Complete the reaction conditions and yield of compound 1. The prodtuct was obtained in small quantity: enough for Elemental analysis and alfa-D, not enough for NMR characterization.

Page 7, line 217: What nitrones were prepared as previously reported? Please add. ADDED

Round 2

Reviewer 2 Report

I now recommend accepting this manuscript for its publication in Molecules

Reviewer 3 Report

The manuscript is publishable.